# Edible Snail Production in Europe

**DOI:** 10.3390/ani12202732

**Published:** 2022-10-11

**Authors:** Anna Rygało-Galewska, Klara Zglińska, Tomasz Niemiec

**Affiliations:** Division of Animal Nutrition, Institute of Animal Sciences, Warsaw University of Life Sciences, Ciszewskiego 8, 02-786 Warsaw, Poland

**Keywords:** edible snails, feed, snail production, mollusc

## Abstract

**Simple Summary:**

Edible snails are a good source of easily digestible nutrients. They are easy to breed and their farming is more environmentally friendly than traditional livestock: they need little space, use less feed per kg of growth, and emit significantly less greenhouse gases. This review aims to present the most important issues related to the breeding of edible snails in European conditions: their importance, maintaining systems, the value of meat and caviar, and the feed used during the animals’ rearing and fattening period.

**Abstract:**

The human population is growing; food production is becoming insufficient, and the growing awareness of the negative impact of traditional animal husbandry on the environment means that the search for alternative methods of providing animal protein is continuously underway. The breeding of edible snails seems to be a promising option. The most popular species of edible snails in Europe include the brown garden snail *Cornu aspersum* (Müller, 1774) (previously divided into two subspecies: *Cornu aspersum aspersum* (Müller, 1774) and *Cornu aspersum maxima* (Taylor, 1883)), as well as the Roman Snail—*Helix pomatia* Linnaeus, 1758. These animals are highly productive, require relatively little space, are easy to breed and their maintenance does not require large financial outlays. This review focuses on the prospects of food snail farming in Europe. It discusses the living conditions, the nutritional value of the snails’ meat, and the way of feeding the animals, paying particular attention to issues still not scientifically resolved, such as the need for micro and macro elements, as well as fat and carbohydrates.

## 1. Introduction

By 2050, the human population is predicted to grow to 9 billion. FAO (Food and Agriculture Organization) reports [1] estimate that food production will increase by 70%. The main challenge for food producers is the production of protein (especially animal protein), the demand for which is still growing. High costs, relatively high water consumption and environmental pollution (mainly by nitrogen and greenhouse gases) deepen the problem of animal protein production [2]. This situation is conducive to implementing innovative, safe technologies of sustainable production or alternative animal husbandry, mainly invertebrates, including molluscs [3,4].

Edible snails are an interesting development opportunity for farmers because they are easy to breed and do not require significant financial outlays to start production, as well as less human labour during the production cycle compared to livestock. They need relatively little space for maintenance, both in the field and indoors. They have good efficiency, emit little greenhouse gases and pollutants into the environment, and are easy to incorporate into organic farming [4,5,6].

Snails are also valued for their low-calorie meat and caviar. In many European coastal countries (Italy, Spain, France), land snails are an integral part of the national cuisine, and the demand for their production is constantly high [7,8].

The most popular species of edible snails in the zone between the Atlantic and Mediterranean parts of Europe include the Brown Garden Snail *Cornu aspersum* (Müller, 1774) (previously divided into two found in two subspecies: *Cornu aspersum aspersum* (Müller, 1774) and *Cornu aspersum maxima* (Taylor, 1883)), as well as the Roman Snail—*Helix pomatia* Linnaeus, 1758. Species of land snails less popular in Europe but consumed elsewhere in the world include: *Archachatina*
*marginata*, *Achatina achatina*, *Achatina fulica* and *Helix lucorum* [9].

Taking the example of *Cornu aspersum*, the snail maintenance cycle lasts 6–7 months and covers the period of waking snails from hibernation (February–March), snail reproduction (at the turn of winter and spring), through hatching and rearing for the first month of snails’ life. The fattening period, in the conditions of Central Europe, lasts from May to the end of September. Then, a small, selected part of the herd is intended for reproduction in the following year, while the remaining part is sold [9]. *Helix pomatia* reaches its maturity after 12–14 months under breeding conditions, and in the wild this time is extended up to four years [10]. One of the reasons for the low profitability of growing it in monoculture is a longer maturation period, and thus a better return on investment due to the lower maintenance of this snail species compared to *Cornu aspersum* [11].

This study aims to collect the available knowledge on the importance of snail production in Europe, housing systems and feeding of these animals.

## 2. The Importance of Industrial Production of Snails

Snails are found in the traditional cuisine of France and Greece, especially in the area of Crete. It is estimated that the domestic European market can cover only 60–70% of the demand for this product. For example, 95% of snails consumed in France (20–40 thousand tonnes) comes from imports [10,12].

Although the beginnings of *Helix pomatia* (Roman snail) breeding go back to 50 B.C. in Rome, they gained popularity again only in the last century, on a large scale, among others, thanks to the development of gastronomy [12,13]. Snail farming has developed significantly in modern times due to the food safety aspects of acquiring wild individuals that bioaccumulate heavy metals in their tissues, including Cd and Pb [14,15], and toxic products of agricultural origin [12,16]. Additionally, in snails raised on farms, fewer *Escherichia coli*, *Enterococcus pp* and *Salmonella* bacteria were detected [17].

According to estimates, in 2016, the consumption of snails in the world amounted to 43,000 tons, and the increase in consumption by 2025 is predicted at 50,000 tons [18]. The largest meat consumption is recorded in Spain, Morocco, France, and Portugal. Morocco played a key role in the producer market in 2020 (15.6% of world exports), Lithuania—8.6% and Romania—7.5%. The largest importers of snails are France (25.3% of world imports), Spain (21.6%) and Romania (8.5%) [19].

Snails, unlike large livestock, need little space and are not as demanding in terms of living conditions or feeding as farm animals. For example, FCR (feed conversion ratio) for *Cornu aspersum aspersum* is 1.3, and for *Cornu aspersum maxima* it is 1.66–1.85 [13,20,21] in the internal housing system and 0.9–1.2 in the mixed system [9]. To compare FCR for cattle, pigs, chicken and fish is 6–10, 2.7–5, 1.61 and 1–2.4, respectively [22,23]. Live weight gain for *Cornu aspersum aspersum* and *maxima* is approximately 2 kg/m^2^. The costs of starting breeding are also low, especially in the case of external keeping. The most significant expenses are in the first year due to the farm preparation for snail keeping [5,6].

A study conducted in southern Italy found the carbon footprint of the *Cornu aspersum maxima* production chain to be 0.7 kg CO_2_ eq per kg of fresh snail meat. In comparison, production of 1 kg of pork, chicken and beef resulted in 3.9–10 kg CO_2_, 3.7–6.9 kg CO_2_ and 14 to 32 kg CO_2_, respectively [24]. Land snail farming also shows significantly lower greenhouse gas production than other traditionally kept livestock. The possibility of binding 0.1 kg CO_2_ in the shell of snails in the form of calcium carbonate for each kilogram of produced snails was also indicated [25]. Additionally, the research by Zucaro et al. [4] showed that snail production has a lower effect on climate change, terrestrial acidification, freshwater eutrophication, marine eutrophication, and photochemical oxidant formation, particulate matter formation and fossil depletion especially compared to beef production.

### 2.1. Meat

Snail meat can be an alternative to the products of other farm animals: it has a low energy value (100 g of meat provides about 60–80 kcal [26]), low content of fat and a high content of exogenous amino acids (*Helix pomatia* 5305 mg/100 g DM (dry matter), *Cornu aspersum aspersum* 4004 mg/100 g DM [27]) and unsaturated fatty acids. Snail meat can be a low-calorie and low-fat source of protein and minerals in the human diet. There are studies showing that snail meat may be an allergen in susceptible individuals, especially in cross-allergies [28,29].

The composition of snail meat may greatly change under the influence of its living conditions (wild vs. farmed) and diet. Among other things, the level and source of protein and fat in the feed are important [30,31,32]. Table 1 and Table 2 show the composition of snail meat; however, the literature on this subject is not very extensive.

### 2.2. Caviar and Mucus Production

Types of edible snail production also include breeding (often carried out together with meat production) and caviar production (known as white caviar, snail eggs or “the pearls of Aphrodite”) started in France and developed in Chile, Italy and Poland. Eggs in the last stage of development or young animals within the first week of hatching are intended for sale for further meat use [12].

Egg production occurs in a mixed housing system from February to the end of April (after the breeding snail is awakened from winter hibernation) or throughout the year in a closed housing system. The eggs are harvested from 4- to 5-month-old *Cornu aspersum* snails. The conditioning of breeders before and during the laying season affects their fertility and the frequency of laying eggs. Breeders should be provided with a properly balanced feed (higher levels of protein and calcium compared to the standard feed) and adequately high humidity and temperature for the breeding season. The eggs are placed in hatching cups filled with soil at 2–3 monthly intervals. The number of eggs laid by breeders in the following months decreases, and their mortality increases [21]. For this reason, breeders are used only once per season, after which they are removed from breeding stock and intended for slaughter [13]. From *Cornu aspersum maxima,* it is possible to get 180 to 250 eggs at a time, while from *Cornu aspersum aspersum*, whose eggs are more preferred for consumption—120 to 200 eggs. The average weight of one packet of one hundred eggs varies from 3 to 6 g, while the diameter of one egg is from 3 to 6 mm. Properly built, fertilised, healthy eggs, which are the raw material for caviar production, have a pearly colour, are opaque and surrounded by a translucent soft shell [11,21,44]. Eggs are collected, washed by hand and individually selected, pasteurised, salted, flavoured, and then packaged in jars [12]. The delicate taste of snail caviar is described as “earthy and nutty” or “mushroom and oak” [45]. It is known from the comparative studies of various types of consumption caviar that grey snail caviar has more fat content than other caviars, and it is almost twice as high as found in sturgeon caviar and amounts to an average of 9.96% DM. On the other hand, the protein content in snail eggs is similar to its level in meat and amounts to approximately 16% as served [46]. Snail eggs are also used in the cosmetic industry—the extracts obtained from them used in creams can effectively reduce skin roughness and dyspigmentation, and improve skin brightness and elasticity [47].

Snail mucus is mainly used in cosmetics—the mucin present in it is particularly valuable as it reduces wrinkles and improves elasticity and hydration of the skin exposed to UVB (Ultraviolet B). At the same time, consuming specifics made from snail mucus does not show any toxic effect on the body [48]. Mucin also has antibacterial properties. Studies have proven its influence on the limitation of the growth *of E. coli*, *Pseudomonas aeruginosa*, *Bacillus subtilis* and *Staphylococcus aureus* [49,50,51] as well as *Streptococcus *sp [52]. It also has been shown that snail mucus has an anti-ageing effect on the skin [53,54] and shows regenerative, anti-inflammatory and wound healing properties [55,56,57]. Its antioxidant properties are also known [58]. In addition, the research of Matusiewicz et al. [59] indicates this natural substance’s anti-cancer effect on colon cancer. Furthermore, snail mucus has been tested as a food packaging material showing high extensibility, high water barrier and antibacterial properties [60].

## 3. Maintaining Snails in the Climate of Central Europe

Three maintaining systems can be distinguished in rearing edible snails: external, internal, and mixed. This chapter concentrates on the *Cornu aspersum* species due to the fact that *Helix pomatia* is still obtained mainly from nature, due to unsatisfactory results of rearing in farms. It is economically justified only to keep Roman snails in common rearing with *Cornu aspersum* [9].

### 3.1. External System

In the external system, snails are kept in parks (plots) throughout development, fattening, breeding and overwintering. This system is popular in countries with a warm climate, such as Italy [13]. It is characterised by a high mortality of young and adult individuals during breeding and overwintering. However, a study carried out in Romania showed that the “sandwich” rearing system (using micro shelters in the field) can protect snails during wintering in the field from adverse weather conditions [61]. Thus, its use can minimise the mortality of animals in external maintenance and may allow the hibernating snails to be safely kept outside. This change could minimise expenses related to purchasing new individuals for the herd and maintaining the cold store.

Plots should not be located in the vicinity of agricultural and forest crops due to the use of plant protection products that may be hazardous to snails. It is also important that the soil is sufficiently loose, rich in humus and calcium, and free from chemical contamination. If possible, it should be protected against drying out, strong wind, flooding and frost. The area should be free of large trees, which could attract predators and cast too much shadow on the plot [9,16,62]. Several to a dozen plant species should be sown, as perennial plants are food for snails (examples can be found below). It is important not to use any chemicals to protect the plants. Instead, it is recommended to introduce natural biological agents, such as marigolds or red clover, which attract predatory mites, lady birds and predatory beetles to help get rid of harmful insects [63].

The recommended stocking density of adult *Cornu aspersum aspersum* is 20–24 individuals per m^2^, *Cornu aspersum maxima* up to 10 individuals per m^2^ and 25 individuals per m^2^ of *Helix pomatia* [16]. The biotic load, in this case, is 250–350 g of the snail mass per m^2^. Low maintenance costs characterise outdoor rearing with perennial plants, and the only technical devices are used for irrigation and protection against pests [21].

The area where the snails are kept should be fenced with a high fence (approximately 2 m), often with corrugated sheet or PVC (Poli Vinyl Chloride), mesh plates, and a net. This way, the snails are protected from strong wind and predators [62,64]. Inside the park, shorter fences divide it into rectangular plots 2 to 3 metres wide, separated from each other by a send path (up to 80 cm wide) that makes the inspection and feeding of snails possible. The height of the internal fencing of the quarters should not exceed 50 cm. This provides easy access to the snails for carrying out all the daily activities related to their care [9,63]. In order to protect snails from escaping, an electrical system with a voltage of 3–5 V, at the height of about 30 cm, can be used. Copper wire (a taste disliked by snails) or two fastened down flaps at the top of the fence can be installed [16,62,65]. Another way of preventing their escape is to attach a cloth tape or string soaked in a mixture of technical petroleum jelly and grey soap to the top of the fence, which repels slugs. A small peak should protect the strap from getting wet [21]. The fence should also be buried to a depth of about 30–40 cm to prevent predators, such as mice and shrews, from digging under it [16]. In order to protect against birds above the park, a net can be hung, with a mesh size preventing the birds from getting inside. The sounds of birds of prey and kites are also used to protect against snail-feeding birds [13,21]. Apart from birds, beetles, lizards, snakes, toads and rats pose a threat to snails as well. Larger animals, such as rabbits, hares and moles, destroy crops and trample animals [13,16,62]. Predator losses can be as high as 60% [21].

Sprinklers are used to irrigate the plots, mostly in the evening and in the morning. Automated systems prefer using water at a temperature close to the air temperature [9,21].

A smaller park area is needed for annual rearing than in long-term farming, relating to the use of perennial plants instead of annual plants, which is more popular. Snails are kept in long quarters with high fences [11]. It is necessary to seed several species of plants in the feeding sequence (weekly sowing intervals) and taller protective plants, reluctantly eaten by snails, to shelter them. Complementary sowing outside the quarters can also be carried out to supplement the diet with fresh green fodder if the plants available in the field are eaten [21,62]. In addition to suitable plant species, permanent places to distribute concentrated feed in troughs sheltered from the weather should be provided. These are also a place of refuge for snails. Feeding and sprinkling should occur in the evening when the snails increase their activity and before dawn [21,66,67].

Under conditions of external rearing, the production cycle begins with the purchase of snail eggs or young animals a few days after hatching. The permissible density for juvenile *Cornu aspersum aspersum* is 100–130 individuals per m^2^; for *Cornu aspersum maxima* 70–100 individuals per m^2^, which gives a biotic load of 1.5–2 kg of adult snails/m^2^ [21,63]. For *Helix pomatia,* the density of individuals reaches 50–150 1-year individuals per m^2^ and biotic load reaches 0.5–1.5 kg/1 m^2^ [11]. The production of snails can also be started with the purchase of adult animals ready for breeding. In spring, when the daily temperature remains above 10 °C, the snails are introduced to the plots at a density of 30–50 individuals per m^2^ (*Cornu aspersum aspersum*) or 20–30 individuals per m^2^ (*Cornu aspersum maxima*). That may occur earlier while using protective devices, such as boxes or tents, guaranteeing a favourable microclimate. Then, the farmer can wait for juvenile snails to appear in the beginning of spring [9]. Feeding animals during conditioning, preceding reproduction by 2–3 weeks, a higher supply of protein (about 18% of the mixture composition) and calcium should be ensured. It results in greater fertility, hatchability of eggs and a higher frequency of egg-laying [21]. Ovipositioning may occur 2 or 3 times at one-month intervals [66]. Individuals who mature early in the summer often reproduce uncontrollably, which is a source of losses related to the specific necessity of their nutrition and increased susceptibility to bacterial diseases. The laying period is expected in the fall; hence early breeding individuals may not be adequately prepared to lay eggs and be appropriately supervised by staff. Reproduction is also associated with higher mortality of snails. In this case, the earlier collection of the individuals should be considered [11].

During the fattening period, from May to September, *Cornu aspersum* snails prefer temperatures between 18–22 °C and moderate humidity. They feed mostly at night and use feed mixtures given on feeding tables. *Helix pomatia* prefers temperatures between 14–20 °C and high humidity. They feed on plants, mostly in the morning [9,13].

The fattening period in Europe ends in autumn at the end of September or the beginning of October when the daily temperature drops below 10 °C. Snails stay on plots from 5 to 7 months [21,66]. At the end of the fattening period, sprinkling and feeding should be reduced. Thanks to these measures, the snails will begin to prepare for hibernation, facilitating their later harvest for trade. The largest specimens with well-developed shells should be intended for keeping the breeding herd and breeding in the spring of the following year. That will ensure the self-sufficiency of snail farming. Breeding hibernation should last at least three months [13,21].

The commercial body weight of *Cornu aspersum aspersum* snails is about 10 g, *Cornu aspersum maxima* about 20 g [21] and Roman snail 12–24 g [11]. The harvest of commodity biomass of 300 individuals/m^2^ density *Cornu aspersum* snails can be expected at the level of 2–4 kg/m^2^ [11]. Individuals intended for trade, since their traditional administration in a dish takes place in the shell, should have a strong and well-curled shell, which proves their maturity and ensures adequate resistance to technological processes related to cleaning, transport and preparation of snails for their consumption [13,21,63].

In this system, snails are kept outdoors for a whole year-long cycle—wintering and rearing hatchlings also take place outside, which increases the risk of deaths associated with this period in the animal’s life [21].

### 3.2. Internal Housing System

Keeping and breeding snails in an internal system is entirely independent of the weather conditions; thus, it is popular in countries with severe winters. This system originated in France [13]. It enables breeding throughout the year but requires financial outlays to construct or adapt a building with adequate thermal insulation and moisture resistance. It should be equipped with running water, sewage, heat monitoring systems, ventilation, overhead sprinklers and appropriate lighting [13]. The rooms should be divided into a cold store (temperature 5 °C and relative humidity of 75%), which is used for hibernation and storage of snails in special boxes, and a reproduction room (with mating tables). It should be equipped with containers filled with soil for laying eggs and with the litter boxes where the eggs are transferred and gently spread over the entire surface with a quill. Both *Cornu aspersum* and *Helix pomatia* eggs should be incubated at a temperature of about 20–22 °C and a humidity of about 80%. Eggs from one reproductor should not be mixed with others due to the uneven hatching time of the young snails and the risk of cannibalism. The eggs of *Cornu aspersum* incubate for about two weeks and of *Helix pomatia* up to three weeks [11,13,21].

Hatcheries are equipped with small boxes for hatchlings, a rearing room and a fattening room. Fodder and technical storage should also be found in the housing system. The walls should be lined with ceramic tiles up to the ceiling and protected against moisture [21,66]. Table 3 presents optimal conditions for snail rearing in a closed system.

For 100 individuals in the basic herd, 1–15 m^2^ of the area should be provided. In the fattening room, it is necessary to reduce the density of individuals by half every 2–3 weeks. In intensive closed fattening, the production cycle lasts for about three months. Three or four cycles can be achieved during the year, thanks to maintaining conditions for reproduction and ovipositioning throughout the year. However, maintenance of this type is expensive mainly due to the constant maintenance of a temperature of about 20 degrees in the rooms for rearing and fattening, as well as the staff handling the animals, and taking care of the hygiene of the rooms and containers for rearing the young snails [21,66].

### 3.3. Mixed Housing System

Currently, the most popular is the mixed system. The mixed system combines the advantages of both of the systems mentioned above. It starts with waking up the hibernating breeders in cold stores, then mating and laying eggs. Rearing young animals takes place in closed rooms. Juvenile snails are kept in rooms until they are 3–4 weeks old and fed with feed mixtures. This increases the survival rate of young snails by reducing pest and weather hazards. Month-old snails also make better use of plant food when transferred to quarters planted with vegetation. They are introduced to the plot at a density of 300–350 individuals per 1 m^2^ for *Cornu aspersum* [9]. For Roman snails, the density reaches 200–400 individuals per 1 m^2^. Too high a density may cause dwarf snails, slight weight gain, health problems and an increase in animal mortality. There, proper fattening takes place [11,13,16]. External maintenance during the fattening of animals reduces the financial outlays and the number of employees necessary to maintain the herd [21,66].

This maintenance system is also used for the combined maintenance of *Cornu aspersum* and *Helix pomatia*. Roman snails should be kept in greenhouses for a year, for the period of their reproduction, as well as growth and maturation of juvenile snails. After the first wintering, which can take place in greenhouse conditions, the snails, together with young *Cornu aspersum*, are transferred to the field plots in spring. In joint maintenance, it is important to sow the plots with more fodder vegetation, as *Helix pomatia* prefers green vegetation over feed mixtures [11]. On the plot, 300 individuals/m^2^ of *Cornu aspersum* are placed along with 15–50 individuals/m^2^ of *Helix pomatia*, which results in harvesting 3 kg/m^2^ of *Cornu aspersum* at the end of fattening period and 0.2–0.5 kg/m^2^ of Roman snail. The harvesting must take place in September due to *Helix pomatia* entering hibernation earlier than the brown garden snail [11].

## 4. Feeding

### 4.1. Digestive System

In the throat of the snail, there is a concholine plate, called a radula, with multiple teeth—it is used to scrape food off the surface. The uncomplicated digestive tract has two salivary glands, and a digestive gland (hepato-pancreas). This gland is also used to store the nutrients absorbed from the intestine. The enzymes found in snails in saliva, gastric juice, stomach, intestine, rectum and the digestive gland include, among others: amylase, protease—cathepsin, trypsin and 1–4,β-polyglucoside, lipase, esterase and cellulose. The gastrointestinal flora of the gastrointestinal tract of the snail plays a vital role in digestion, with its composition similar to the environment in which the snail lives. The digestive system ends with the rectum next to the breathing opening [21,68,69].

### 4.2. Feeding in External Conditions

Plants growing in quarters where snails are kept are the basis of their diet in external maintenance. Complete feed applied between rows of vegetation supplements the ration with an additional portion of concentrated nutrients and energy, especially protein and calcium. Table 4 presents the results of numerous studies on the food preferences of the snails *Cornu aspersum* and *Helix pomatia*.

Chevalier et al. [74] show that snails avoid bitter plants and prefer plants rich in calcium and carbohydrates, especially glucose and fructose [75].

### 4.3. Feed Mixtures

Feed mixtures are the basis for feeding snails in internal maintenance and supplementing the diet in external and mixed maintenance.

The basis of concentrated feed for snails is cereal meal (corn, barley, oat), wheat bran, soybean and the addition of a source of calcium (most often limestone), dried green fodder, oil and a vitamin-mineral mixture [11,21].

Three phases characterise the development of snails—slow growth in the first month (when the development of internal organs occurs), 2–3 months of rapid weight gain, followed by stabilisation of body weight and entering the period of sexual maturity and reproduction. Each of these stages is characterised by different requirements for the composition of the feed. A greater supply of calcium and mineral mixture is used during breeding, and most commonly, the mineral mixture for laying hens is used (Table 5). For juveniles, a mineral mixture for chickens is added, while during fattening, the protein content of the feed increases and mineral mixtures for rearing pigs are used [9,11,76]. Mineral additives help keep animals healthy at various stages of development and positively affect the development and growth of the organism.

The extent of the contribution of nutrients to the feed used in snail feed is shown in Table 6. Examples of feed mixtures used in feeding land edible snails are shown in Table 7.

To reduce the cost of feed production, 10% of soybean meal can be replaced with peameal, beans, and sweet lupine. Due to snails’ poor digestion of gluten, it is not recommended to use a large amount of wheat [81].

Studies on *Cornu aspersum* snails have shown protein digestibility at 65.1%, fat at 82.7% and calcium at 49.2% [32]. It was estimated that the feed consumption per 1 kg of feed mixtures results in 1.3 kg snail weight gain in *Cornu aspersum* and 1.5 kg in *Helix pomatia* [21].

**Table 7 animals-12-02732-t007:** Examples of feed mixtures used in feeding edible landedible snails.

Components	%	Snail Species	Reference
Wheat bran	35.00	*Cornu aspersum aspersum*	[21]
Barley flour	35.00		
Dried carrot flour	10.0		
Mineral concentrate	20.00		
Soybean flour	16.00	*Cornu aspersum aspersum*	[34]
Corn gluten	11.00		
Wheat flour	28.00		
Canola oil	3.00		
Dicalcium phosphate	5.00		
Pectin	1.00		
Limestone	33.00		
Sodium chloride	0.50		
Vitamin premix	1.50		
Mineral premix	1.00		
Maise	54.00	*Archachatina marginata*	[82]
Brewer’s spent grain	22.00		
Soybean	8.00		
Groundnut	7.00		
Fish meal	5.00		
Oyster shell	2.00		
Bone meal	1.50		
Premix	0.50		
Maise	60.00	*Achatina achatina*	[83]
Wheat bran	10.00		
Soybean meal	10.00		
Russia fish	9.00		
Tuna	8.00		
Oyster shell	25.00		
Premix	0.50		

### 4.4. Protein

Zhou et al. [84] found that protein requirements are a priority in nutritional research. Protein is the highest cost in commercial feed formulation and plays a vital role in animal growth; insufficient protein in the diet results in growth restriction, arrest, and weight loss. Protein provides the amino acids for building body tissues and allows for maximum growth; if too much is supplied in the diet, only some of it is used to produce new protein in the body, and the rest is converted into energy. That results in an increased and unnecessary feed cost.

Based on Polish studies [85] conducted on *Cornu aspersum* snails, it was established that the protein from soybean meal as its primary source in the feed at 18.6% (vs. 16.7%) increased the diameter of the shells of individuals kept in greenhouse pens. In the case of *Cornu aspersum aspersum* maintained externally, the increased protein content affected the carcass weight of the snails. The percentage of leg in carcass weight and body weight was higher in snails fed with a mixture with a lower proportion of protein [85].

Other teams in their experiments showed that *Cornu aspersum aspersum* snails utilise the feed best and gain weight when protein makes up 10% of the feed composition [86]. In contrast, Sampelayo et al. [87] showed that optimal protein retention and weight gain were achieved with a feed content of 17.5%.

Studies conducted on *Cornu aspersum maxima* showed that the optimal protein level in the feed is 18%. Both lower and higher (21%) protein levels resulted in lower weight gain for snails. However, the 18% level had the lowest animal survival rate (92%) compared to other study groups. Individuals in this group showed the smallest proportion of shell weight in overall body weight. The protein sources used in this study were soybean meal, methionine, and lysine combined [37].

For the *Cornu aspersum maxima*, the optimal protein level was set at 16%, with soybean meal as its source [88].

Studies on the *Archachatina*
*marginata*, which also belongs to edible snails, more prevalent in African countries, have shown a beneficial effect of groundnut cake and fish meal as a source of protein in feeding on weight gain and length of snail shells [82]. Nyameasem and Borketey-La [83] established that growing protein content from soybean meal in feed increased feed intake by this species, as well as their final body weight and shell width. A higher amount of protein (with soybean meal as its source) in the feed also resulted in lower FCR. It was established that 22% protein in feed results in significantly higher body mass of snail feed intake, FCR, shell length, and width [89].

Wacker and Baur [90], during an experiment on the species *Ariunta*
*arbustorum*, showed that high casein-derived protein levels in snails’ diet (21.3%) had a positive effect on weight gain, time of reaching sexual maturity and the final size of animals.

### 4.5. Fat

Zhou et al. [91] showed in their experiment that dietary lipids play a significant role in providing concentrated energy, essential fatty acids, phospholipids, sterols, and fat-soluble vitamins, especially for carnivorous snails. It has been shown that increasing the digestible energy content of snail diets through lipid supplementation has a protein-sparing effect, thus reducing nitrogen losses to the environment. Another study [92] showed that providing adequate energy with dietary lipids can minimise the use of more expensive protein as an energy source.

Milinsk et al. [38] analysed the effect of adding 3% vegetable oils to *Cornu aspersum maxima* feed on the production results of snails. The highest final body weight was obtained with the addition of hemp oil (7.69 g) and the lowest with soybean oil (6.5 g). Sunflower oil yielded the lowest carcass yield (28.47%), while rice oil yielded the highest (33.33%). In the fresh weight of snail meat, the lowest percentage of ash (1.01%) was recorded in the group fed with rapeseed oil and the highest with rice oil (1.23%). The highest protein content was recorded in the group fed with soybean oil (18.4%) and the lowest in the group fed with sunflower oil (14.8%). The fat percentage was highest in individuals from the group fed with rice oil (1.29%) and the lowest in the group fed with soybean oil (0.91%). The most favourable ratio of n6/n3 fatty acids was obtained in the group fed with linseed oil (5.01), the least favourable in the group fed with corn oil (7.05). The lowest FCR was achieved with corn oil (1.66) and the highest with soybean oil (1.84). Using hemp and sunflower oils resulted in the lowest survival rate of snails (87.5%), while corn oil completely prevented falls during the experiment.

A study on the snail species *Arianta*
*arbustorum* showed that low cholesterol and PUFA fatty acids in the diet affected lower feed consumption and increased mortality in these animals. The low proportion of PUFA fatty acids also contributed to a lower mating frequency because of the essential role in regulating the reproduction of snails [20].

### 4.6. Energy

The nutrients supplied with the ration are sufficient for maintaining the animal’s essential life functions (metabolism) and for its growth, maturation, reproduction and movement.

The diet of land snails is based on plant food, and their energy requirement is lower than that of discussed aquatic gastropods, which are predominantly carnivorous.

The energy level found in feed for edible land snails is as follows: for *Cornu aspersum aspersum* from 2500 [93] to 3150 kcal/kg [30], for *Cornu aspersum maxima* about 2500 kcal/kg [38,88] and *Helix pomatia* 2200–2400 kcal/kg [94].

In feeding the African snail *Archachatina*
*marginata*, the range of energy concentration in the feed rations is from 2200 kcal/kg up to 4300 kcal/kg [78,95,96].

### 4.7. Calcium

Regarding snail feeding, the research subject is both the source of calcium added to the feed and its share in the feed mixture at different stages of snail development. An adequate calcium level in the diet is fundamental during the juvenile period when snails grow intensively and strengthen their shells. During maturity and reproduction, calcium plays an essential role in egg formation. Deficiencies of this element in the diet lead to a significant increase in the mortality rate of snails during this period of life.

Calcium is the main building block of snail shells [97]. Snail shells contain from 95% to 99% calcium carbonate from calcite and argonite crystals. The organic substance in the shell is primarily conchiolin—a protein specific to molluscs [98]. A hard and durable shell is essential for the processing industry because of its automation and large-scale production. Soft shells can be destroyed and thus lose the possibility of further use at the initial production stage. A soft shell prevents mechanical cleaning and sorting of shells and transports over long distances, increasing processing costs and stopping the development of the industry. The quality of the shells also influences the assessment of the value of the snails intended for export since they serve dishes prepared according to traditional French recipes [99].

Nutrition significantly impacts the calcium content of the shell and, as a result, its properties. With a lower calcium content in the diet of land snail *Achatina fulica*, the content of this element in the mature shell was significantly higher. In contrast, the excess calcium in the feed led to inhibition of snail growth and thinning of the shell [100]. The author explains this phenomenon by the harmful effect of metals, an admixture of calcium carbonate, which weaken the structure of the shell by displacing calcium from it. A balanced calcium level in the feed increased the thickness of the shell. Voelker [101] noted that calcium deficiencies lead to shading the shell and, consequently, greater damage susceptibility. The effect of calcium levels on the dimensions of the shell and its thickness has also been proven [79,102,103].

Both calcium deficiency and its excess in the diet may result in a slower increase in the animal’s body weight [100,104]. The calcium level in feed corresponding to the animal’s demand for this element increases weight gain [102,105]. Excess calcium in the diet can adversely affect the morphology and histochemistry of the digestive tubules of the digestive system of snails [106].

The need for calcium is most significant in adults laying eggs due to its large mobilisation from the organism (especially the shell, which acts as a storehouse of calcium) to produce egg casings and create stocks of this element in the bodies of young snails [100,107]. About 65% of this element in the snail’s body during ovipositioning is mobilised [108]. The final amount of calcium in the diet of snails is conducive to improving the reproductive results of these animals—the number of eggs laid, the hardness of eggshells, hatchability, and survival of juveniles increases [103,109,110].

The excessive outflow of mineral substances from the shell during the intensive period of reproduction and laying of eggs often leads to the death of snails or the occurrence of cannibalism. Providing adequate calcium supplementation increases the survival rate of adults and young adults in the first month of life [95,100,105].

Depending on the species of snail and the source of calcium in the feed, the optimal range of the share of the source of this element in the feed mixture is in the range of 16–40%. Among other things, the use of limestone (most commonly used in ready fodder feed), bone meal, eggshells, and snail shells were subjected to the diets of growing snails [102]. A study by Aman et al. [103] achieved the best production results using the last source of calcium. In a study on the *Archachatina*
*marginata* for 20 months after hatching, the best results were achieved using oyster shells or cattle bone meal at 30% of the ratio. On the same species in the same period, an experiment was carried out using Mikhart as a source of calcium. The best results were obtained at 30% of this calcium source in the feed. The 40% level showed significant decreases in production results [104]. Juveniles of the *Archachatina* species were fed plants with the addition of a source of calcium from eggshells, limestone, wood ash, oyster shells and bone meal. Levels of 10, 20, 30 and 40% of the diets dry matter were used as a calcium source. The best production results were achieved with 20% oyster shells [111,112]. Ikauniece et al. [113] recommend using 20% pure calcium carbonate for juvenile snails. It contains about 40–55% calcium in the composition and small amounts of sodium, magnesium, and iron [114]. Furthermore, a study on *Cornu aspersum aspersum* showed that the level of calcium carbonate in feed at 22.5% gives better results than 12.5% during the juvenile stage of snail development [105].

Studies carried out by Wacker Wacker and Baur [90] on the species *Ariuntu*
*arbustorum* has shown that snails try to compensate for calcium deficiencies in feed by increasing calcium intake. However, in the case of feeds with low and medium levels of this element, they cannot take up enough calcium for the body’s regular metabolism and the growth of shells. Calcium deficiency in this study was also associated with higher mortality of snails. Acidic soils, poor in calcium, lead to cannibalism in snails (shell overeating) as a result of the search for a source of calcium [115].

### 4.8. Feed Additives

Feed additives are used in animal nutrition to improve their health and production performance. However, little research on feed additives in the diet of snails has so far been carried out.

Studies on the addition of betaine in 5 g/kg of feed for *Cornu aspersum aspersum* snails kept in laboratory conditions have significantly affected animal growth up to 10 weeks of age [116].

Ligaszewski and Pol [117] showed that in the feeding of *Cornu aspersum maxima* that were externally maintained, the addition of garlic preparation or its combination with probiotics (live lactobacilli) resulted in lower sails mortality and higher commercial biomass. The use of an antibiotic additive adversely affected all production results.

A study was also conducted in outdoor farming to examine the effect of painting feed tables for *Cornu aspersum aspersum* with silver nanoparticles and Multimicrobial Preparation (EM) on hygienic conditions and production results. Both additives had a positive effect on the survival of the animals. The addition of EM positively affected the body weight and the concentration of Mg and Fe in snail carcasses. The experiment also established that the fat content of snail meat increased, along with the crushing strength of the shell [33,118].

## 5. Conclusions

In the era of the growing demand for animal protein and, at the same time, the introduction of restrictions on animal breeding for environmental reasons, snails seem to be a promising direction of development. They are undemanding in terms of workload, financial outlays and space requirements, their production causes little environmental burden, and they have good production efficiency. The feeding of snails is based on feeds similar in composition to those used for farm animals. At the same time, snails are characterised by good feed utilisation for weight gain and do not require high large financial outlays for feed.

In snail maintenance research, many factors have been established and optimised. When it comes to feeding these animals, the effects of protein and calcium levels and sources are best understood. Still, a less-investigated area is the influence of sources and levels of fat in the diet, levels and sources of carbohydrates, especially simple sugars and dietary fibre. Additionally, little is known about the effects of mineral and vitamin supplements. The nutrition of edible land snails is particularly noteworthy because they are popular in Western Europe for culinary use.

Further and in-depth research on the nutrients mentioned above is suggested for closed-farm mollusc development, health, dietary and technological properties.

## Figures and Tables

**Table 1 animals-12-02732-t001:** The mean composition values of various snail meats (% of DM) and the composition of the main group of fatty acids in the meat of edible snails (% share in total fat).

	*Cornu aspersum aspersum*		*Cornu aspersum maxima*		*Helix pomatia*	
	Mean	SD	Mean	SD	Mean	SD
Crude protein	67.42	11.69	59.53	10.84	80.17	8.15
Crude fat	4.77	3.04	7.30	4.90	4.24	2.39
Crude ash	8.38	5.19	4.98	2.17	9.48	3.24
SFA *	24.99	4.50	24.38	4.16	32.40	9.17
MUFA **	26.39	5.85	22.82	3.73	18.00	3.74
PUFA ***	45.83	9.54	52.39	7.95	37.34	18.78
n6/n3 ****	6.33	4.65	7.79	3.10	5.87	2.81
Reference	[27,30,31,32,33,34,35]		[32,35,36,37,38]		[32,35,36,39,40,41]	

SD (standard deviation). * Saturated Fatty Acids. ** Monounsaturated Fatty Acids. *** Polyunsaturated Fatty Acids. **** PUFAs Ratio

**Table 2 animals-12-02732-t002:** The mean share of elements and vitamins in the meat of edible snails.

(mg/100 g)		*Cornu aspersum*					*Helix pomatia*				
Na	91.95	-	-	-	-	-	336.54	90.50	-	-	-
Ca	135.7	-	-	3277.50	1620.00	-	48.08	726.25	4580.00	-	-
K	105.4	-	-	-	-	-	1836.54	82.17	-	-	-
Mg	17.05	-	320.37	-	425.00	-	1201.92	54.05	375.00	-	-
P	96.72	-	1285.10	634.75	590.00	-	1307.69	104.52	1049.00	-	-
Fe	0.52	-	22.41	11.37	8.40	-	16.83	1.71	1.10	-	-
Zn	-	-	7.11	4.12	8.00	-	1.00	1.35	8.80	-	-
Vit A	5.46	-	-	-	-	-	0.14	-	-	-	-
Vit E	0.88	-	-	-	-	-	24.04	-	-	-	-
Vit C	-	-	-	-	-	18.54	-	-	-	3.81	-
Vit B1	0.15	-	-	-	-	-	0.05	-	-	-	-
Vit B2	0.07	-	-	-	-	-	0.58	-	-	-	-
Vit B3	3.23	-	-	-	-	-	6.73	-	-	-	-
Vit B6	0.29	-	-	-	-	-	0.63	-	-	-	-
Amino acids (% DM)											
Val	0.72	2.33	-	-	-	-	-	-	-	-	0.38
Leu	0.61	3.96	-	-	-	-	-	-	-	-	0.50
Ile	0.47	1.87	-	-	-	-	-	-	-	-	0.31
Phe	0.36	2.06	-	-	-	-	-	-	-	-	0.42
Lys	0.72	2.01	-	-	-	-	-	-	-	-	0.41
Thr	0.45	2.94	-	-	-	-	-	-	-	-	0.35
Met	0.43	0.78	-	-	-	-	-	-	-	-	0.11
Trp	-	1.80	-	-	-	-	-	-	-	-	0.85
His	0.25	0.89	-	-	-	-	-	-	-	-	0.24
Reference	[27]	[30]	[33]	[34]	[36]	[42]	[12]	[41]	[36]	[42]	[43]

**Table 3 animals-12-02732-t003:** Optimal conditions for snail rearing in a closed system [9,11,21].

	Value	
	Min–Max	Optimal
Lighting	30–200 lux	100 lux
Duration of the light cycle		-
One month	12 h	
From the second month	18 h	
Relative humidity	65–90%	90% ± 5
Temperature:		20 °C ± 2
-life cycle	10–30 °C	
-reproduction	18–22 °C	
-incubation	22–24 °C	
-hibernation	0–7 °C	

**Table 4 animals-12-02732-t004:** Plant species preferred by edible snails.

		Reference
Herbs and weeds	apple mint (*Mentha suaveolens*) *Brassica rapa var. sylvestris* burdock (*Arctium*) clover (*Trifolium praténse L., Trifolium repens*) courgette (*Cucurbita pepo*) creeping buttercup (*Ramunculus* *repens*) lucerne (*Medicago*) sorrel (*Rumex acetosa*) stinging nettle (*Urtica dioica*) white mustard (*Sinapis alba*)	[7,10,11,13,70,71,72,73]
Grass	orchard grass (*Dactylis glomerata*) red fescue (*Festuca rubra*) soft brome (*Bromus hordeaceus*)	
Vegetables	artichoke (*Cynara scolymus*) bean (*Phaseolus*) cabbage (*Brassica*) carrot (*Daucus carota*) cauliflower (*Brassica oleracea var. botrytis*) chicory (*Cichorium intybus*) cucumber (*Cucumis sativus*) garden pea (*Pisum*) lettuce (*Lactuca* *sativa*) oilseed rape (*Brassica napus*) pumpkin (*Cucurbita moschata*) radish (*Raphanus*) sugar beet (*Beta vulgaris*) tomato (*Solanum lycopersicum*) turnip (*Brassica rapa subsp. rapa*) turnip rape (*Brassica rapa ssp. oleifera*)	
Cereals	barley (*Hordeum*) oat (*Avena*) wheat (*Triticum*)	
Flowers	blue violet (*Viola sororia*)	

**Table 5 animals-12-02732-t005:** The composition of exemplary vitamin and mineral mixtures used for snails.

Mixture	Vitamin and Mineral Mixture Used during Fattening *	Vitamin and Mineral Mixture Used during Breeding **
Composition in 1 kg of mixture	Biotin—880 mcg Nicotinic acid—226.4 mg Pantothenic acid 17.7 mg Folic acid—8.8 mg Iron—153.6 mg Choline—2300 mg Cystine—1.8 g Threonine—11.8 g Tryptophan—2.2 g Isoleucine—11 g Leucine—15 g Tyrosine—21 g Phenylalanine—23.4 g Valine—11.7 g Histidine—8.6 g Arginine—12.9 g Proline—4.5 g Asparagine 21 g Lysine—17.9 g Methionine—3.8 g Calcium—195 g Phosphorus—6.3 g Sodium—8 g Vitamin A—6.4 IU Vitamin E—3 mg Vitamin B1—6.6 mg Vitamin B2—19.9 mg Vitamin B6—13.2 mg Vitamin B12—20 mcg	Niacinamide 300 mg Pantothenic acid 118 mg Folic Acid 10 mg Iron 900 mg Manganese 1200 mg Copper 90 mg Zinc 1000 mg Iodine 10 mg Selenium 2 mg Calcium 290 g Phosphorus 20 g Magnesium 4 g Sodium 14 g Vitamin A 150,000 IU Vitamin D3 30,000 IU Vitamin E 200 mg

* mixture used as a dietary supplement for pigs. ** mixture used as a dietary supplement for laying hens.

**Table 6 animals-12-02732-t006:** The proportion of nutrients used in feed mixtures for edible land snails.

	% DM	Reference
Crude protein	10.00–33.33	[21,34]
Crude fat	2.34–6.00	[77,78]
Crude ash	7.50–40.56	[78,79]
Crude fibre	5.00–7.85	[21,79]
Carbohydrates	22.26	[34]
Calcium	6.00–20.90	[77,80]
Energy (kcal/kg)	2100–3200	[9,31]

## Data Availability

Not applicable.

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
