# Peer review of "Edible Snail Production in Europe"

_animals, 2022, doi:10.3390/ani12202732_

Round 1

Reviewer 1 Report

Dear authors, 

You attempted to review the the most important issues related to the breeding of edible snails in Europe.

I find your work to have a lot of potential however after you revise several parts of it. 

Please see my comments on the manuscript. 

Pay special attention to the cohesion of your text and the correct use or absence of citations!

Author Response

Dear Reviewer,
We would like to thank you for valuable comments and efforts towards improving our manuscript. As this is my first publication, as the first author, I would also like to thank you very much for your in-depth and developed comments of great value to me. I learned a lot thanks to your comments.

Please see the attachment, I added responses to your comments on .pdf version of manuscript.

Reviewer 2 Report

The authors submitted a literary review containing various information concerning the farming of edible terrestrial snails in Europe. As such, this publication can be accepted if it fits the aims and scope of the journal. However, its scientific significance is not evident to me. I have added many specific comments and questions directly to the pdf version of the text. Also, I ask the authors to use the comments given below. 

1.       My first concern about this manuscript is that the target audience of it remains unclear. It hardly can be interested for zoologists, whereas the snail farmers can take only most basic facts from it (I am not sure that the snail farmers read Animals journal). In other words, the manuscript looks much more as a textbook chapter (or even a user’s manual) than a scientific publication. However, I do not dare to judge on the suitability of this text for the journal; it is a task of the Editorial Board.

Though the authors discuss two species of popular edible snails in Europe (i.e., Cornu aspersum and Helix pomatia), the content of their article is devoted chiefly to one of them (C. aspersum). For example, section 3 contains information only on the two “varieties” of Cornu aspersum, without even a single mention of Helix pomatia. A reader can learn too small about the Helix pomatia farming from this review. In my opinion, the authors must correct this bias and to add more data on the Roman snail

Author Response

Dear Reviewer,
We would like to thank you for valuable comments and efforts towards improving our manuscript. As this is my first publication, as the first author, I would also like to thank you very much for your in-depth and developed comments of great value to me. I learned a lot, thanks to your comments.

I would like to answer to your main comments here:

  1. My first concern about this manuscript is that the target audience of it remains unclear. It hardly can be interested for zoologists, whereas the snail farmers can take only most basic facts from it (I am not sure that the snail farmers read Animals journal). In other words, the manuscript looks much more as a textbook chapter (or even a user’s manual) than a scientific publication. However, I do not dare to judge on the suitability of this text for the journal; it is a task of the Editorial Board.

-Thank you for this comment. We wrote this publication for people planning to conduct research on edible snails in field conditions. When planning our research, it was very difficult to find all the reliable and research-based knowledge in one place (basically there is no such publication according to our research), especially regarding the nutrition of these animals. Hence, the audience of our publication are mainly scientists planning research on snails and breeders who take their breeding business seriously. We believe that when searching for information about edible snail production, they will also find their way to this publication.

Though the authors discuss two species of popular edible snails in Europe (i.e., Cornu aspersum and Helix pomatia), the content of their article is devoted chiefly to one of them (C. aspersum). For example, section 3 contains information only on the two “varieties” of Cornu aspersum, without even a single mention of Helix pomatia. A reader can learn too small about the Helix pomatia farming from this review. In my opinion, the authors must correct this bias and to add more data on the Roman snail

- this issue has been addressed and developed in a new version of the manuscript.

I replied to the rest of your comments directly in .pdf version of the manuscript - please see the attachment.

Round 2

Reviewer 1 Report

Dear authors, 

you have extensively revised your work. I would suggest a careful read through in order to correct potential spelling mistakes. For example you write CO2 and not CO. Otherwise the anuscript has significantly improved. I have no more comments to add.

Reviewer 2 Report

I must thank the authors for their extensive work on the improvement of the original version of the MS. As I could see, most of my criticisms and suggestions have been taken into account by the authors. The new version of the paper is significantly better than the original one. I think it can be accepted for publication without further revision(s)